# Effects of Supplementation of 25-Hydroxyvitamin D$_3$ as a Vitamin D$_3$ Substitute on Performance, Bone Traits, and Egg Quality of Laying Hens from 1 Day to 72 Weeks of Age

**Dongdong Li** [1,2], **Xuemei Ding** [1], **Shiping Bai** [1], **Jianping Wang** [1], **Qiufeng Zeng** [1], **Huanwei Peng** [1], **Yue Xuan** [1] **and Keying Zhang** [1,*]

[1] Animal Nutrition Institute, Key Laboratory of Animal Disease-Resistance Nutrition, Ministry of Education, Ministry of Agriculture and Rural Affairs, Key Laboratory of Sichuan Province, Sichuan Agricultural University, Chengdu 611130, China
[2] College of Animal Science, Xichang University, Xichang 615000, China
* Correspondence: zkeying@sicau.edu.cn; Tel.: +86-28-8629-0922

**Abstract:** This experiment was conducted to explore the effect of long-term supplementation of 25-hydroxyvitamin D$_3$ (25-OHD) as a vitamin D$_3$ (VD$_3$) substitute on performance, bone traits, and egg quality of laying hens from 1 day to 72 weeks of age. In total, 900 one-day-old Lohman pullets were randomly allotted into three dietary groups (three treatments × 15 replicates × 20 birds per replicate): VD$_3$ 2800 IU/kg; 25-OHD 69 µg/kg; 25-OHD 125 µg/kg. At the end of the 20th w, five replicates from each group were selected to feed on the same vitamin D diets, as used during the rearing stage (1–20 w) until 72 w. The result showed that the 25-OHD 125 µg/kg treatment had the lowest average daily feed intake (ADFI) at 1–8 or 1–19 w, body weight at 8 w, body weight gain between 1 and 8 w and shank length at 4 w ($p < 0.05$). The 25-OHD 125 µg/kg treatment had a lower shank length at 7 w, compared with the 25-OHD 69 µg/kg treatment. The shank length of the birds in each treatment reached the maximum (about 103 mm) at about 18 w of age. For the bone traits, the 25-OHD 125 µg/kg treatment had the lowest femur bone diameter at 20 w ($p < 0.001$) and femur bone plumpness at 20 w ($p = 0.002$). The 25-OHD 125 µg/kg treatment had a lower tibia strength at 10 w ($p = 0.023$) and keel length at 10 w ($p = 0.046$), compared with the 25-OHD 69 µg/kg treatment. However, both 25-OHD 69 and 125 µg/kg treatments had a greater femur strength at 72 w ($p = 0.006$), compared with the VD$_3$ 2800 IU/kg treatment. No difference in laying performance was observed among all treatments. The overall (21–72 w) ADFI in the 25-OHD 125 µg/kg treatment was significantly lower than that in the 25-OHD 69 µg/kg treatment ($p = 0.030$). At 60 w, the 25-OHD 125 µg/kg treatment had a lower eggshell thickness ($p = 0.012$) and proportion of eggshell ($p = 0.022$), compared with the 25-OHD 69 µg/kg treatment. No significant differences in egg quality parameters were observed at 50 and 70 w among treatments. In general, supplementary 2800 IU/kg doses of VD$_3$ at the early stage were sufficient to maintain the bone quality and growth and development of pullets. Feeding birds at a higher 25-OHD level (125 µg/kg) resulted in the reduced ADFI and growth at the rearing period, but the long-term supplementation of 25-OHD as a VD$_3$ substitute improved the bone quality in the late laying period.

**Keywords:** 25-hydroxyvitamin D$_3$; bone traits; egg quality; laying performance; laying hen

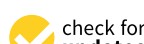



## 1. Introduction

Modern commercial laying hens are more productive and have a longer laying cycle. This is a huge challenge for the health of the hen's various tissues and organs, specifically in bone health. For hens, bone quality and eggshell quality are closely related. Medullary bone can provide 40% of the calcium in the eggshell during daily eggshell formation [1]. Furthermore, some studies have shown that keel fractures in birds had a negative effect on

the laying performance and egg quality [2,3]. Thus, mineral reservoirs, such as Ca in the bones, are important for maintaining healthy bones and the best eggshell quality.

In the poultry industry, vitamin $D_3$ ($VD_3$) has long been the most commonly used supplemental form of vitamin D in diets. Since 2006, 25-hydroxyvitamin $D_3$ (25-OHD), a metabolite of $VD_3$, has been allowed as an additional form of dietary vitamin D in the poultry industry [4,5]. Compared with $VD_3$, 25-OHD had a higher absorption efficiency and stronger affinity with vitamin D binding protein [6]. Due to functional defect and weakening of the liver and kidney in young pullets and old laying hens [7,8], the use of 25-OHD in the laying hen diet can be a good choice.

Vitamin D can maintain the homeostasis of blood calcium by regulating intestine calcium absorption, kidney calcium reabsorption, and bone mineralization and mobilization [9,10]. Vitamin D is essential for maintaining the laying performance, egg quality, and bone quality of laying hens [11]. Studies have shown that partial and complete replacement of $VD_3$ with 25-OHD in diets improved the eggshell quality [12–14] and the yolk color [15], and reduced dirty and broken egg rates [16] in breeder hens or laying hens. Otherwise, a previous study found that adding 25-OHD to a basal diet containing $VD_3$ increased the thickness of eggshells [17]. However, a lower daily feed intake and egg weight were reported, when $VD_3$ was completely substituted for 25OHD in laying hen diets [18]. Numerous studies have shown that the dietary addition of 25-OHD did not significantly improve the eggshell quality and egg production performance [5,19–21]. For bone health, the positive effects of 25-OHD on bone quality in laying hens were not observed in most of studies [5,6,12,19,20]. A study also showed that 25-OHD had no effect on the prevalence of keel deformity [22]. However, A recent study showed that dietary 25-OHD significantly increased the content of medullary ashes and decreased the concentration of cortical ashes in the femur and tibia [23]. There is a big difference between these findings. This may be related to the age of the laying hens, duration of the experiment, and the total vitamin D activity in the diet. Taken together, most of studies do not cover the pullet period and a complete cycle in laying hens. A recent study found that early and long-term supplementation of 25-OHD significantly improved the laying performance and egg quality during early laying period, when $VD_3$ was substituted for 25-OHD at 50%, in the birds' diets, at 5520 IU/kg of feed of the total vitamin D activity [8]. However, so far, there has been no report on the effect of 25-OHD on the performance, the quality of the egg, and bone traits of hens, when 25-OHD completely replaces dietary $VD_3$ for a complete cycle in laying hens.

Hence, the purpose of the this experiment was to study the effects of 25-OHD as a $VD_3$ substitute on hens' growth and bone development, laying performance, egg quality, and bone traits from 1 day to 72 weeks of age.

## 2. Material and Methods

### 2.1. Birds, Housing, and Treatments

All processes were permitted by the Animal Care and Use Committee of the Sichuan Agricultural University (SAUPN-19-02).

Initially, on the basis of the principle of no difference in body weight, a total of nine hundred one-day-old Roman Pink laying hens (three treatments × 15 replicates × 20 birds per replicate) were randomly distributed into three dietary treatments: $VD_3$ 2800 IU/kg, 25-OHD 69 μg/kg and 25-OHD 125 μg/kg. Starting from 21 w, five replicate birds from each treatment were selected continued to feed on the same vitamin D diet as during the rearing period (1–20 w) and up to 72 w.

The experiment was conducted in a closed poultry house, and the laying hens were raised in cages with dimensions measuring 400 mm in length, 450 mm in width, and 450 mm in height. The chicken cage is 192 cm wide, 62.5 cm deep and 57 cm high. There was no restriction on chicken drinking and feeding throughout the experiment. Seven basic diets were assigned to seven stages (0–4, 5–8, 9–17, 18–19, 20–45, 46–65, 66–72 w) from 1 day to 72 weeks of age. the basic diet composition and nutrient levels at each stage are shown

in Table 1. The environmental conditions were controlled according to 'Roman Commercial Layer Management Guide (2018)'. The ambient temperature of the pullets at 1 day of age was controlled at 35–36 °C. Then the temperature gradually decreased to 20 °C at 25 days of age. The pullets were given 24 h of light at 1 day of age, then the light time gradually decreased, and the illumination time decreased to 8 h at 8 w. When hens were 18 weeks of age, light was supplemented to stimulate laying eggs until 14 h. At 24 w, light was then kept the same until the end of the experiment.

**Table 1.** Composition and nutrient level of the basal diet.

| Ingredient (%) | 0–4 w | 5–8 w | 9–17 w | 18–19 w | 20–45 w | 46–65 w | 66–72 w |
|---|---|---|---|---|---|---|---|
| Corn | 59.92 | 63.98 | 68.34 | 62.42 | 57.00 | 58.02 | 60.58 |
| Soybean meal | 34.90 | 31.19 | 17.80 | 29.06 | 28.36 | 26.65 | 24.26 |
| Wheat bran | | | 9.22 | | | | |
| Soybean oil | 0.44 | 0.52 | 0.75 | 1.50 | 3.24 | 3.24 | 2.93 |
| DL-methionine | 0.21 | 0.11 | 0.17 | 0.18 | 0.29 | 0.26 | 0.23 |
| L-lysine HCL | 0.17 | 0.03 | | | | | |
| L-tryptophan | | | 0.01 | 0.02 | | | |
| L-threonine | 0.04 | | | | | | |
| NaCl | 0.18 | 0.17 | 0.16 | 0.16 | 0.16 | 0.16 | 0.16 |
| Choline chloride, 60% | 0.05 | 0.05 | 0.05 | 0.07 | 0.07 | 0.07 | 0.07 |
| NaHCO$_3$ | | | | | 0.25 | 0.25 | 0.25 |
| Calcium carbonate | 1.27 | 1.26 | 1.36 | 3.90 | 8.67 | 9.42 | 9.75 |
| Calcium hydrophosphate | 2.08 | 1.95 | 1.40 | 1.95 | 1.57 | 1.54 | 1.38 |
| Mineral premix [1] | 0.5 | 0.5 | 0.5 | 0.5 | 0.15 | 0.15 | 0.15 |
| Vitamin premix [2] | 0.23 | 0.23 | 0.23 | 0.23 | 0.23 | 0.23 | 0.23 |
| Antioxidant (ethoxyquin) | 0.01 | 0.01 | 0.01 | 0.01 | 0.01 | 0.01 | 0.01 |
| Total | 100 | 100 | 100 | 100 | 100 | 100 | 100 |
| Calculated nutrient content, % | | | | | | | |
| Metabolizable Energy (kcal/kg) | 2753 | 2789 | 2783 | 2775 | 2738 | 2729 | 2725 |
| Crude protein | 20.00 | 18.50 | 14.50 | 17.50 | 16.82 | 16.15 | 15.30 |
| Calcium | 1.05 | 1.00 | 0.90 | 2.00 | 3.73 | 4.00 | 4.09 |
| Non-phytate P | 0.48 | 0.45 | 0.37 | 0.45 | 0.38 | 0.37 | 0.35 |
| Lysine | 1.20 | 1.00 | 0.68 | 0.92 | 0.88 | 0.84 | 0.78 |
| Methionine | 0.51 | 0.40 | 0.39 | 0.45 | 0.55 | 0.51 | 0.47 |
| Tryptophan | 0.23 | 0.21 | 0.17 | 0.22 | 0.19 | 0.18 | 0.17 |
| Threonine | 0.80 | 0.71 | 0.54 | 0.67 | 0.65 | 0.62 | 0.59 |

[1] Provided per kilogram of diet (0–72 w): Cu (CuSO$_4$·5H$_2$O) 5 mg, Fe (FeSO$_4$·H$_2$O) 25 mg, Mn (MnSO$_4$·H$_2$O) 100 mg, Zn (ZnSO$_4$·H$_2$O) 60 mg, I (KI) 0.5 mg, Se (Na$_2$SeO$_3$) 0.2 mg. [2] Provided per kilogram of diet (0–8, 9–17, 18–72 w): 10,000, 10,000, 10,000 IU vitamin A; 30, 30, 30 mg vitamin E; 3, 3, 3 mg vitamin K$_3$; 1, 1, 1 mg vitamin B$_1$; 6, 6, 4 mg vitamin B$_2$; 3, 3, 3 mg vitamin B$_6$; 20, 20, 25 μg vitamin B$_{12}$; 8, 8, 10 mg D-pantothenate; 30, 30, 30 mg niacin acid; 1, 1, 0.5 mg folic acid; 50, 50, 50 μg biotin. The three levels of vitamin D VD$_3$ 2800 IU/kg, 25-OHD 69 μg/kg and 25-OHD 125 μg/kg were added to the seven diets used, respectively.

## 2.2. Data Collection and Sampling

Feed intake at each stage (1–8, 9–19, 1–19 w) and hen body weight at each time point (1 d, 8 w and 19 w) were recorded, and average daily feed intake and body weight gain were calculated. Body weight uniformity (BWU) of each replicate at 8 w and 19 w was calculated. BWU (%) = number of hens within the range of 10% add and subtract the birds average weight/ the number of all hens in the duplicate. At 1, 2, 3, 4, 5, 6, 7, 8, 10, 13, 16, and 19 w, the shank length of the hens (select 10 birds from each replicate) was measured and recorded, and the measurement of the shank length was measured with a vernier caliper. Starting from 21 w, egg production performance data were recorded daily. Then, the laying rate, average egg weight, feed conversion, feed intake, qualified egg number, total egg number, and total egg weight at the laying stage were calculated. At 10, 20, and 72 w, the birds were euthanized, and the full keel bones and left and right tibias and femurs were collected, and the keel bones were stored at 4 °C until measured, The femurs and tibias are stored at −20 °C. Chickens sampled at weeks 10 and 20 were selected in the first

five replicates of each treatment. Then, after 20 w, the middle five replicates were uniformly selected to continue the experiment.

### 2.3. Egg Quality

At 50, 60, and 70 w, three eggs were selected from each replicate to determine the egg quality. Egg quality parameters included the Haugh unit, albumen height, eggshell thickness, eggshell breaking strength, and proportion of eggshell. Haugh unit and albumen height were determined by a multifunctional egg analyzer (EMT-7, 300, Robotmation Co., Ltd., Tokyo, Japan). Eggshell strength was tested using an intensity measuring device (Robotmation Co., Ltd.). Using a vernier caliper to measure the big end, middle end and small end of the eggshell.

### 2.4. Bone Development and Bone Strength

At 10 and 20 w, after stripping off the left and right tibia and femur, the bone length and diameter (the middle of the bone) of tibias and femurs were tested with a vernier caliper, and bone plumpness was calculated. Bone plumpness (%) = bone diameter × 3.14/bone length × 100%. Then, bone strength was measured by the texture analyzer (TAXTPlus, Stable MicroSystems corp., Godalming, England). At 72 w, after stripping out the intact tibias and femurs, the bone strength was measured.

At 10 and 20 w, after separating the intact keel bones, the parameters (keel depth, keel length, and keel calcified rate) related to keel development and calcification were measured. The sites for the determination of the keel development and calcification parameters refer to the description of my previous article [24].

### 2.5. Statistical Analysis

All data were analyzed statistically by one-way ANOVA, using the general linear model procedure of SAS (version 9.1) with dietary treatment as the main effect. Duncan's method was used for multiple comparisons. When $p$ was less than 0.05, the difference was significant. Otherwise, a quadratic regression was used to predict the bird shank length for each treatment using SPSS software (version 19) with the following model:

$$Y = AX2 + BX + C$$

where Y is the shank length, X is weeks of age, and C is the intercept of the equation, B is the linear term coefficient, and A is the quadratic term coefficient. The coefficient of determination ($R^2$) and $p$-value regression were used to define the equation with the best fit. A $p$-value less than 0.05 was considered statistically significant.

## 3. Results

### 3.1. Growth Performance and Shank Length

The 25-OHD 125 μg/kg treatment group had the lowest BW (8 w), BWG (1–8 w), and ADFI (1–8 w and 1–19 w) among all treatments ($p < 0.05$; Table 2). No significant difference was found in BW (1 d), BWG (9–19 w), ADFI (9–19 w), BW (19 w), BWG (1–19 w), and BWU (8 w and 19 w).

The 25-OHD 125 μg/kg treatment group had the lowest shank length among all treatments at 4 w ($p = 0.027$; Figure 1). The 25-OHD 69 μg/kg treatment group showed a higher shank length, compared with the 25-OHD 125 μg/kg treatment group at 7 w ($p = 0.032$). However, no significant difference in the shank length was observed at 1, 2, 3, 5, 6, 8, 10, 13, 16, and 19 w ($p > 0.05$). By a regression analysis, we found the shank length of the birds in each treatment group reached the maximum (about 103 mm) at about 18 w (Table 3).

**Table 2.** Effect of 25-OHD as a VD3 substitute on hens' growth performance [1].

| Items | VD$_3$ 2800 IU/kg | 25-OHD 69 µg/kg | 25-OHD 125 µg/kg | SEM | *p*-Value |
|---|---|---|---|---|---|
| BW, g | | | | | |
| 1 d | 41.7 | 41.7 | 41.7 | 0.1 | 0.994 |
| 8 w | 666.4 [a] | 669.2 [a] | 644.4 [b] | 5.2 | 0.003 |
| 19 w | 1624.5 | 1633.8 | 1611.2 | 10.6 | 0.326 |
| BWG, g | | | | | |
| 1–8 w | 624.7 [a] | 627.5 [a] | 602.8 [b] | 5.2 | 0.003 |
| 9–19 w | 958.2 | 964.5 | 966.8 | 11 | 0.851 |
| 1–19 w | 1582.9 | 1592 | 1569.5 | 10.6 | 0.326 |
| BWU, % | | | | | |
| 8 w | 76.5 | 77.7 | 68.8 | 3.0 | 0.083 |
| 19 w | 67.4 | 71.8 | 69.6 | 3.0 | 0.578 |
| ADFI, g | | | | | |
| 1–8 w | 30.5 [a] | 30.4 [a] | 29.7 [b] | 0.2 | 0.001 |
| 9–19 w | 72.6 | 72.3 | 71.9 | 0.4 | 0.504 |
| 1–19 w | 54.6 [a] | 54.4 [a] | 53.9 [b] | 0.2 | 0.014 |

[a,b] Means with different superscripts within a row differ significantly ($p < 0.05$). Each mean repre–ents values from 15 replicates. [1] Abbreviations: BW, body weight; BWG, body weight gain; BWU, body weight uniformity; SEM, standard error of the mean.

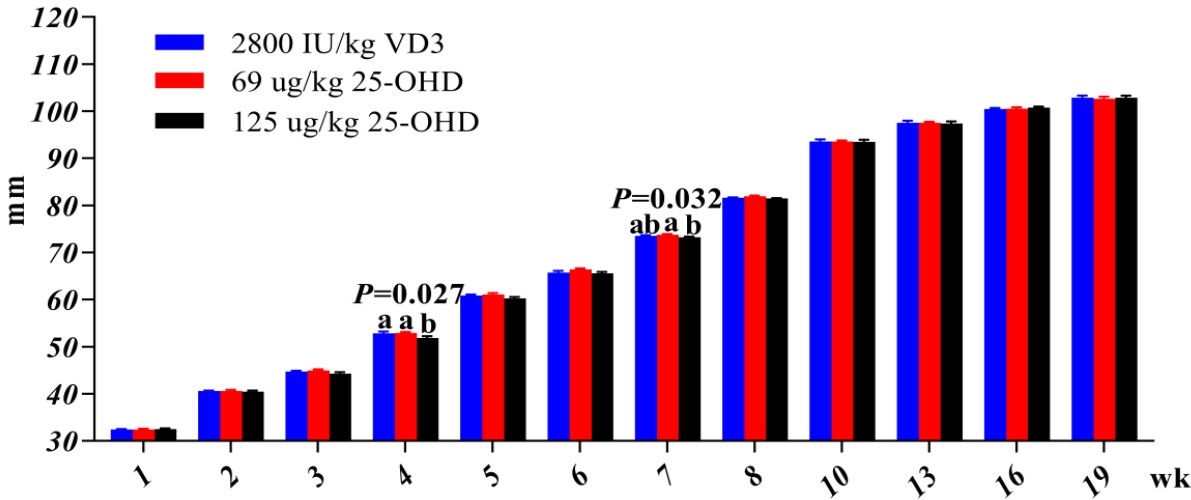

**Figure 1.** Effect of 25-OHD as a VD3 substitute on hens' shank length; SEM = standard error of the mean; each mean represents values from 15 replicates. a,b means significant difference ($p < 0.05$).

**Table 3.** Regression equations for the prediction of the shank length from the week age of laying hens [1].

| Treatment | Prediction Equations | R$^2$ | *p*-Value | Y (Max) | X |
|---|---|---|---|---|---|
| VD$_3$ 2800 IU/kg | Y= −0.278 X$^2$ + 9.554 X + 20.983 | 0.989 | <0.001 | 103.1 | 17.2 |
| 25-OHD 69 µg/kg | Y= −0.281 X$^2$ + 9.599 X + 21.043 | 0.991 | <0.001 | 103.2 | 17.1 |
| 25-OHD 125 µg/kg | Y = −0.275 X$^2$ + 9.529 X + 20.715 | 0.988 | <0.001 | 103.3 | 17.3 |

[1] R$^2$, the coefficient of determination; Y, shank length; X, week age of laying hens; max, maximum.

### 3.2. Bone Development and Bone Quality

Bone development results are presented in Table 4. The 25-OHD 125 µg/kg treatment group had the lowest femur bone diameter ($p < 0.001$; 20 w) and femur bone plumpness ($p = 0.002$; 20 w) among all treatments. The 25-OHD 125 µg/kg treatment group had a

lower keel length ($p$ = 0.046; 10 w), compared with the 25-OHD 69 µg/kg treatment group. There were no significant differences in the tibia length, tibia diameter, tibia plumpness, femur length, and keel depth at 10 and 20 w among treatments ($p$ > 0.05).

**Table 4.** Effect of 25-OHD as a VD3 substitute on bone development at 10 and 20 weeks.

| Items | VD$_3$ 2800 IU/kg | 25-OHD 69 µg/kg | 25-OHD 125 µg/kg | SEM | $p$-Value |
|---|---|---|---|---|---|
| 10 w | | | | | |
| Tibia length, mm | 98.9 | 101.5 | 97.2 | 1.3 | 0.112 |
| Tibia diameter, mm | 6.1 | 6.2 | 5.7 | 0.2 | 0.167 |
| Tibia plumpness, % | 19.2 | 19.2 | 18.5 | 0.5 | 0.587 |
| Femur length, mm | 69.7 | 70.1 | 69.8 | 0.8 | 0.945 |
| Femur diameter, mm | 6.7 | 6.6 | 6.6 | 0.1 | 0.701 |
| Femur plumpness, % | 30.3 | 29.6 | 29.8 | 0.5 | 0.681 |
| Keel length, mm | 76.0 [ab] | 79.4 [a] | 72.7 [b] | 1.7 | 0.046 |
| Keel depth, mm | 30.1 | 30.3 | 30.2 | 0.8 | 0.982 |
| 20 w | | | | | |
| Tibia length, mm | 116.9 | 116 | 116.1 | 0.8 | 0.684 |
| Tibia diameter, mm | 7.2 | 7.3 | 7.2 | 0.1 | 0.231 |
| Tibia plumpness, % | 19.3 | 19.9 | 19.4 | 0.2 | 0.136 |
| Femur length, mm | 80.9 | 77.8 | 78.8 | 1.1 | 0.159 |
| Femur diameter, mm | 7.7 [b] | 7.9 [b] | 7.0 [a] | 0.1 | <0.001 |
| Femur plumpness, % | 30.0 [b] | 31.9 [c] | 28.1 [a] | 0.5 | 0.002 |
| Keel length, mm | 104.6 | 105.4 | 102.9 | 1.4 | 0.470 |
| Keel depth, mm | 39.5 | 40.6 | 39.5 | 0.7 | 0.379 |

[a–c] Means with different superscripts within a row are significantly different ($p$ < 0.05). Each mean represents values from five replicates.

As shown in Table 5, the 25-OHD 125 µg/kg treatment group had a lower tibia strength at 10 w. compared with the 25-OHD 69 µg/kg treatment group. ($p$ > 0.05), but both 25-OHD 69 and 125 µg/kg treatment groups had a higher femur strength at 72 w, compared with the VD$_3$ 2800 IU/kg treatment group. No significant difference in the keel calcified rate was observed at 10 and 20 w ($p$ > 0.05).

**Table 5.** Effect of long-term supplementation of 25-OHD as a vitamin D3 substitute on bone strength and keel calcification.

| Item | VD$_3$ 2800 IU/kg | 25-OHD 69 µg/kg | 25-OHD 125 µg/kg | SEM | $p$-Value |
|---|---|---|---|---|---|
| 10 w | | | | | |
| Tibia strength, kgf | 15.1 [ab] | 17.4 [b] | 12.8 [a] | 0.9 | 0.023 |
| Femur strength, kgf | 21.0 | 21.9 | 21.7 | 1.3 | 0.872 |
| Keel calcified rate, % | 38.6 | 39.0 | 39.5 | 1.7 | 0.932 |
| 20 w | | | | | |
| Tibia strength, kgf | 17.0 | 17.2 | 19.4 | 1.5 | 0.488 |
| Femur strength, kgf | 25.1 | 27.0 | 23.8 | 2.6 | 0.692 |
| Keel calcified rate, % | 85.3 | 86.6 | 86.9 | 1.7 | 0.789 |
| 72 w | | | | | |
| Tibia strength, kgf | 16.0 | 18.3 | 18.2 | 1.1 | 0.262 |
| Femur strength, kgf | 23.9 [a] | 34.0 [b] | 31.5 [b] | 1.7 | 0.006 |

[a,b] Means with different superscripts within a row are significantly different ($p$ < 0.05). Each mean represents values from five replicates.

### 3.3. Laying Performance and Egg Quality

As can be seen from Figure 2, the hen-day laying rate (HDLR) in each treatment reached peak production (more than 90%) at 23 w. Prior to 50 w, the HDLR was greater than 90% in each treatment group, then the HDLR gradually declined. At 72 w, the egg production rate of each treatment group was about 85%. As shown in Table 6, no significant difference was also found in the laying performance among all treatments except for the

feed intake. The ADFI in the 25-OHD 125 µg/kg treatment group was significantly lower than that in the 25-OHD 69 µg/kg treatment group ($p$ = 0.030).

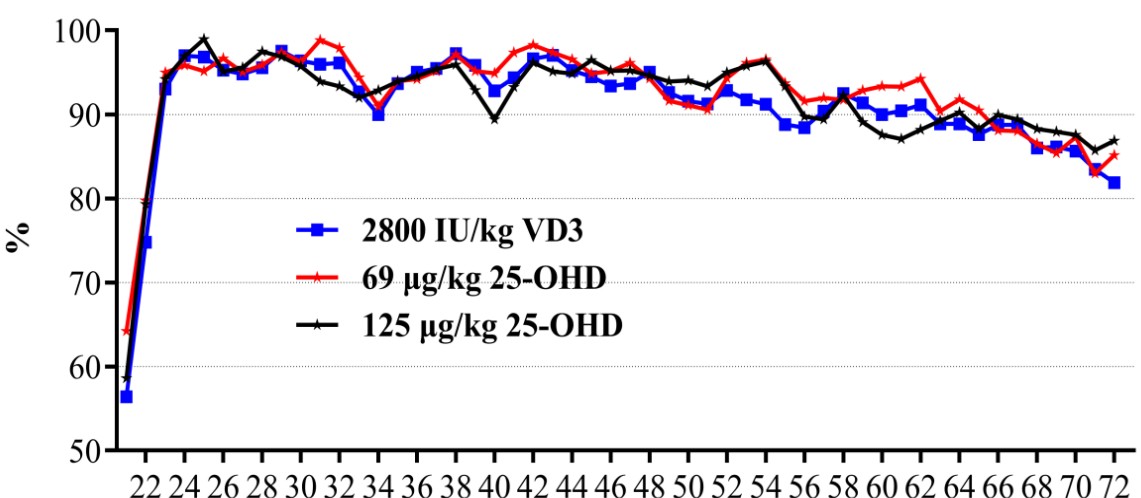

**Figure 2.** Effects of 25-OHD as a VD$_3$ substitute on the laying rate of the hens. Each mean represents values from five replicates.

**Table 6.** Effect of long-term supplementation of 25-OHD as a VD$_3$ substitute on the laying performance and mortality rate during the production stage [1].

| Item | VD$_3$ 2800 IU/kg | 25-OHD 69 µg/kg | 25-OHD 125 µg/kg | SEM | *p*-Value |
|---|---|---|---|---|---|
| Mortality rate, % | 9.0 | 9.3 | 14.4 | 3.5 | 0.484 |
| HDLR, % | 91.6 | 93.3 | 92.3 | 0.5 | 0.147 |
| HHLR, % | 89.2 | 88.0 | 87.3 | 1.8 | 0.790 |
| AEW, g | 62.9 | 62.4 | 62.5 | 0.4 | 0.597 |
| ENHD, No | 330 | 336 | 332 | 2 | 0.184 |
| ENHH, No | 321 | 317 | 314 | 7 | 0.783 |
| HDEW, kg | 20.8 | 21.0 | 20.8 | 0.2 | 0.689 |
| HHEW, kg | 20.2 | 19.8 | 19.7 | 0.4 | 0.684 |
| ADFI, g | 115.5 [ab] | 116.1 [a] | 114.3 [b] | 0.1 | 0.030 |
| FCR | 2.00 | 2.00 | 1.98 | 0.01 | 0.485 |

[a,b] Means with different superscripts within a row are significantly different ($p$ < 0.05; $n$ = 5). The number of hens-housed was calculated as the actual number of birds at 21 w. [1] Abbreviations: 25-OHD, 25-hydroxycholecalciferol; HDLR, hen-day laying rate; HHLR, hen-housed laying rate; AEW, The average egg weight; ENHD, egg number per hen-day; ENHH, egg number per hen-housed; HDEW, hen-day total egg weight; HHEW, hen-housed total egg weight; ADFI, average daily feed intake; FCR, feed conversion ratio.

As shown in Table 7, at 60 w, the 25-OHD 125 µg/kg treatment group had a lower eggshell thickness ($p$ = 0.012) and proportion of eggshell ($p$ = 0.022), compared with the 69 µg/kg 25-OHD treatment group. No significant differences in egg quality parameters were observed at 50 and 70 w.

**Table 7.** Effect of the long-term supplementation of 25-OHD as a VD3 substitute on egg quality at 50, 60, and 70 w.

| Items | VD$_3$ 2800 IU/kg | 25-OHD 69 μg/kg | 25-OHD 125 μg/kg | SEM | *p*-Value |
|---|---|---|---|---|---|
| 50 w | | | | | |
| Eggshell strength, kg/cm$^2$ | 4.62 | 4.24 | 4.4 | 0.15 | 0.233 |
| Eggshell thickness, mm | 0.417 | 0.399 | 0.397 | 0.008 | 0.234 |
| Proportion of eggshell, % | 11.3 | 11 | 11 | 0.2 | 0.616 |
| Albumen height, mm | 7.2 | 6.8 | 6.6 | 0.3 | 0.321 |
| Haugh unit | 82 | 79.8 | 78.9 | 2 | 0.533 |
| 60 w | | | | | |
| Eggshell strength, kg/cm$^2$ | 4.38 | 4.5 | 3.99 | 0.17 | 0.113 |
| Eggshell thickness, mm | 0.386 [ab] | 0.397 [a] | 0.373 [b] | 0.005 | 0.012 |
| Proportion of eggshell, % | 11.2 [ab] | 11.5 [a] | 10.9 [b] | 0.1 | 0.022 |
| Albumen height, mm | 6.6 | 6.7 | 6.4 | 0.3 | 0.832 |
| Haugh unit | 78.7 | 79.3 | 77 | 2.3 | 0.763 |
| 70 w | | | | | |
| Eggshell strength, kg/cm$^2$ | 3.99 | 4.11 | 4.17 | 0.14 | 0.653 |
| Eggshell thickness, mm | 0.389 | 0.381 | 0.378 | 0.008 | 0.607 |
| Proportion of eggshell, % | 10.9 | 10.8 | 10.8 | 0.2 | 0.880 |
| Albumen height, mm | 6.7 | 6 | 6.4 | 0.3 | 0.300 |
| Haugh unit | 79.3 | 74 | 77.2 | 2.1 | 0.232 |

[a,b] Means with different superscripts within a row are significantly different ($p < 0.05$; $n = 5$).

## 4. Discussion

In this experiment, the body weight of the birds in all treatments reached the standard of the 'Roman Commercial Layer Management Guide (2018)' at 8 and 19 w. However, in the starter period (1–8 w), the BWG and ADFI in the 25-OHD 125 μg/kg treatment group was the lowest among all treatments. Previous studies showed that adding 25-OHD to a VD$_3$-containing basal diet promoted bird growth and development up to 3 weeks of age, increasing the feed intake and body weight gain and reduced the FCR [6]. Otherwise, when laying hens were fed three vitamin D diets (VD$_3$ 2760 IU/kg, VD$_3$ 5520 IU/kg, and VD$_3$ 2760 IU/kg + 25OHD 69 μg/kg) for a long period of time (0–95 w), there was no significant difference in the growth performance during the rearing period [8]. These results are inconsistent with ours, which may be caused by inconsistent experimental design. In our experiment, 25-OHD was used to completely replace VD$_3$ in the diet. In actual production, we hardly found VD$_3$ poisoning in laying hens. Earlier studies showed that dietary VD$_3$ 15,000 IU/kg had no negative effect on the laying performance of Roman LSL White laying hens aged 20 to 68 weeks [25]. One study also showed that no poisoning was observed, when Hy-Line W36 laying hens (19–58 w) were fed VD$_3$ 102,200 IU/kg [26]. However, a more recent study found that long-term (0–68 w) supplementation of VD$_3$ 68,348 IU/kg had adverse impacts on the growth performance and egg production [27]. So far, there has been no report on the toxic dose of 25-OHD in laying hens. It has been reported that kidney production of 24,25 dihydroxycholecalciferol is 7–9 times higher than that of 1,25(OH)$_2$D$_3$ in growing pullets [28], however, at the onset of laying, compared with 25-OHD 24-hydroxylase, the 25-OHD 1α-hydroxylase activity on the birds' kidneys, plasma levels of 1,25(OH)$_2$D$_3$, and contents of intestinal 1,25(OH)$_2$D$_3$ were improved significantly [29,30]. This suggested that immature pullets may have lower vitamin D requirements, compared with sexually mature laying hens. In China, the limit level for VD$_3$ used in poultry is 5000 IU/kg. In this experiment, compared with the VD$_3$ 2800 IU/kg treatment, the growth performance of pullets in the 25-OHD 125 μg/kg treatment group was not improved. The optimal dose of 25-OHD for pullets at the rearing stage needs further research.

The bird's shank length is closely related to the skeletal development and frame size [31,32]. Previous studies have shown that hens fed a diet supplemented with 25-OHD have a longer shank length at 18 w, compared with the VD$_3$ 3000 IU/kg treatment [6].

However, in our research, in addition to the growth performance, we also found that the shank length at 4 w in the 25-OHD 125 μg/kg group were the lowest among all treatments, which was associated with a lower feed intake at 1–8 w.

Similar to our results, a previous study found that supplementing with 25-OHD in the early (0–17 w) and long-term (0–90 w), had no effects on the cumulative egg production and egg weight of Hy-Line Brown hens from 18 to 87 w [6]. A recent report showed that early-term and the long addition of 25-OHD had no effect on egg production throughout the period (22–95 w) [8]. Furthermore, In addition, some studies on laying hens and broiler breeders also showed that the addition of 25-OHD to the diet had no beneficial effect on the laying performance [4,5,12,19,20,33]. In these reports, supplementation of 25-OHD was only in the laying stage. In our experiment, we found that the long-term (1–72 w) supplementation of 25-OHD as a $VD_3$ substitute did not affect the laying performance (HDLR, HHLR, AEW, ENHD, ENHH, HDEW, HHEW) of laying hens. This may be attributed to the fact that the total activity of vitamin D and the level of calcium in all treatment diets meet the requirements of laying hens, the addition of $VD_3$ and 25-OHD on top of this do not further improve production performance. One study also showed that the laying performance was significantly reduced when birds were fed a $VD_3$-free diet, however, there was no significant difference in egg production at $VD_3$ supplemental levels of 500, 1500, and 3000 IU/kg [34].

In this study, long-term supplementation of 25-OHD as a vitamin $D_3$ substitute had no beneficial effects on the eggshell quality at 50, 60, and 70 w. The 25-OHD 125 μg/kg treatment had a lower eggshell thickness and proportion of eggshell, compared with the 69 μg/kg 25-OHD treatment group. There has always been controversy about the effect of 25-OHD on eggshell quality. Some studies have shown that 25-OHD had no effect on eggshell quality [4,8,18–20]. However, other studies showed that dietary supplementation of 25-OHD or replacing $VD_3$ with 25-OHD improved the shell quality [12,17]. Silva (2017) also concluded that dietary supplementation of 25-OHD during the rearing and early laying period improved the eggshell thickness [6]. A recent study also found that the dietary supplementation of high levels of $VD_3$ or 25-OHD (125 μg/kg) improved the laying performance and eggshell quality of laying hens, compared to a control group (62.5 μg/kg $VD_3$) [35]. The mechanism of vitamin D in eggshell formation has not been well established. The $1,25(OH)_2D_3$, as an active metabolite of vitamin D, that may regulate the expression of $Ca^2+$ transport-related proteins, such as calcium-binding protein d28k and carbonic anhydrase in the eggshell gland [36]. The production of $1,25(OH)_2D_3$ in the kidneys is related to blood calcium levels. The decrease of blood calcium leads to the increase of parathyroid hormone secretion, which increases the production of $1,25(OH)_2D_3$ in the kidneys [1]. Therefore, the effect of vitamin D on the eggshell quality is inextricably related to dietary calcium levels. In our experiment, dietary calcium levels were adequate for the birds, this may be one of the reasons why 25-OHD as a $VD_3$ substitute did not further enhance the eggshell quality.

The bone quality of birds not only affects the laying performance and egg quality, but is also related to the welfare of laying hens [37,38]. The modern layers have a high incidence of osteoporosis at the later laying period [39,40]. Bone growth and development are concentrated in the early stages, so it is necessary to improve bone quality as much as possible during this stage. Studies on the effects of 25-OHD on bone development in pullets are limited. In our experiment, supplementation of 25-OHD as a $VD_3$ substitute had no beneficial effect on the bone development and bone strength (tibia and femur) at 10 and 20 w. The 25-OHD 125 IU/kg treatment was the worst among all treatments at 10 and 20 w, which may be associated with a lower feed intake and body weight. However, my previous research found increasing dietary $VD_3$ (2800 vs. 300 IU/kg) or the addition of 25-OHD 56 IU/kg during the pullet period improved the tibia quality during the early and later stages [24]. The difference in the results between the two studies was related to the level of VD3 in the control diets in the experiment. On the basis of my previous study, this study further demonstrated that $VD_3$ 2800 IU/kg in hens' diet in the rearing period was sufficient

for the maintenance of bone quality. The results of this study were also inconsistent with the findings in broilers, one study showed that supplementing with 25-OHD via water, reduced the incidence of lameness [41]. One study found that supplementation of 25-OHD increased the bone mineralization and improved bone strength in broilers [42]. For laying hens in the laying stage, most of studies found that 25-OHD did not improve the bone quality [5,12,19,20]. In this study, the long-term addition of 25-OHD significantly improved the strength of the femur at 72 w. This indicates that the beneficial effect of 25-OHD on the bone quality in the late laying period is not caused by early supplementation, but due to continuous supply during the laying period. A recent study also found that the long-term supplementation with 25-OHD stimulated bone growth and had positive effects on laying hen's bone quality [43].

In the laying hen industry, keel bone damage has become a major concern. Keel bone damage has negative impacts on production in laying hens [2]. Studying the regularity of keel bone development and calcification may provide new ideas for reducing keel damage. In this study, supplementation of 25-OHD as a $VD_3$ substitute did not affect the development and calcification of the keel. There is no report on keel development and calcification in pullets. A recent study reported that dietary supplementation with 25-OHD significantly increased the sternal mineral accumulation in meat ducks [44]. A previous study found that the dietary addition of 25-OHD had no effect on the incidence of keel deformity [22]. Whether 25-OHD can be used to improve keel quality is worth further study.

### 5. Conclusions

The study confirmed that the dietary supplementation of $VD_3$ 2800 IU/kg at the early stage (1–20 w) was sufficient to maintain the bone quality and growth of the pullet. The long-term supplementation of 25-OHD as a $VD_3$ (125 μg/kg) substitute improved the bone quality in the late laying period.

**Author Contributions:** Conceptualization, D.L., X.D. and K.Z.; data curation, D.L.; formal analysis, D.L.; funding acquisition, X.D. and K.Z.; methodology, D.L., X.D., S.B., J.W., Q.Z., H.P. and K.Z.; project administration, Y.X.; writing—original draft, D.L.; writing—review and editing, X.D. and K.Z. All authors have read and agreed to the published version of the manuscript.

**Funding:** This study was supported by the National Key Research and Development Program (2016YFD0501202) and Sichuan Provincial Key Research and Development Program (2018NZ0009).

**Institutional Review Board Statement:** The study was approved by the Animal Care and Use Committee of the Sichuan Agricultural University (SAUPN-19-02).

**Data Availability Statement:** Some or all data, models, or code that support the findings of this study are available from the corresponding author upon reasonable request.

**Acknowledgments:** Thanks to Sichuan Shengdile Village Ecological Food Co., Ltd. for their support of this study.

**Conflicts of Interest:** This article does not involve any conflict of interest.

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
