# Peer review of "Effects of Supplementation of 25-Hydroxyvitamin D3 as a Vitamin D3 Substitute on Performance, Bone Traits, and Egg Quality of Laying Hens from 1 Day to 72 Weeks of Age"

_agriculture, doi:10.3390/agriculture13020383_

Round 1

Reviewer 1 Report

The article submitted for review “Effects of supplementation of 25-hydroxyvitamin D3 as a vitamin D3 substitute on performance, bone traits, and egg quality of laying hens from 1 day to 72 weeks of agecontains some interesting information and can be accepted for printing in Agriculture Journal after making the following corrections:

In the Abstract section, there is no room to use abbreviated forms.

L: 21-22 – please reworded because according to Figure 1 the 25-OHD 125 μg/kg treatment had lower shank length only compared with the 25-OHD 69 μg/kg treatment.

L: - 22-23 – ‘The shank length of the birds in each treatment reached the maximum (about 103 mm) at about 18 wk of age.’ - no such data available in the results. Furthermore, in the methodology (L: 116-117), the Authors state that the shrank length of hen was measured and recorded at 1, 2, 3, 4, 5, 6, 7, 8, 10, 13, 16 and 19 wk.

L: 23-25 – Wrong conclusion. In the case of tibia strength and keel length at 10 wk, there are no significant differences between VD3 2800 IU/kg and 25-OHD 125 μg/kg treatment. (see Table 4 and 5).

L: 58 – ‘York color’? Moreover, spell out the full term of VD3 when using for the first time.

L: 87 – Please change ‘×20’ to ‘× 20’

L: 93 – change ‘( 0–4, …)’ to ‘(0–4, …)’

L: 123 – change ‘4 ° C until measured,’ to ‘4 °C until measured.’

L: 157 – 158: please complete because according to Table 2 no significant differences were found also in BW (1d), BWG (9-19wk), and ADFI (9-19wk).

L: 162 – Please reword because no significant differences in shank length were also observed at 1, 2, 3, 5, 6, 8, 10, 13, and 16 weeks (see Figure 1).

Table 4 – (heading of the table) ‘1’ - lack of explanation. Moreover, according to methodology (L: 134- 137 - ‘At 10, 20 and 72 wk, after stripping off the left and right tibia and femur, bone length and diameter (the middle of the bone) of tibia and femur were tested with vernier caliper, and bone plumpness was calculated. Bone plumpness (%) = bone diameter × 3.14 / bone length × 100%.’) bone development and strength were analysed at 10, 20 and 72 weeks. Question: where are the data from week 72?

L: 180-181 - This is not correct conclusion. Consistent with Table 5, there are no significant differences in tibia strength at 10wk between VD3 2800 IU/kg and 25-OHD 125 μg/kg treatment.

Table 6 – Please clarify - 'ENHD - egg number per hen-day.' Question: per one hen?

Reviewer 2 Report

Dear,

Your paper is very similar with two previously published papers. You must include those papers in your work in discussion part.

Citation: Jing, X.; Wang, Y.; Song, F.; Xu, X.; Liu, M.; Wei, Y.; Zhu, H.; Liu,Y.; Wei, J.; Xu, X. A Comparison between Vitamin D3 and 25-Hydroxyvitamin D3 on Laying Performance, Eggshell Quality and Ultrastructure, and Plasma Calcium Levels in Late Period Laying Hens. Animals 2022, 12, 2824. https://doi.org/10.3390/ani12202824

Citation: Li, D.; Zhang, K.; Bai, S.;Wang, J.; Zeng, Q.; Peng, H.; Su, Z.;Xuan, Y.; Qi, S.; Ding, X. Effect of25-Hydroxycholecalciferol withDifferent Vitamin D3 Levels in theHens Diet in the Rearing Period onGrowth Performance, Bone Quality,Egg Production, and Eggshell Quality. Agriculture 2021, 11, 698.

Reviewer 4 Report

Dear Authors,

I have some comments that have shown below: authors should do it 

Line 15: Depend of the total name it is better to write (25-HOD) instead of 25-OHD!

Line 18: Change prelicate to replicate!

Line 18: Change 20 wk to 20th wk.

Line 58: Change York to Yolk 

Line 90: It is not clear! whether authors have control group from each treatment or just one control group? Authors must clear it.

Line 101: What about rearing system is it on floor or in cages? Authors should mention.

Line 166; Table 2, The significant letters should be superscript.

Line 184: Table 4, While there is superscript number in the title, it should be explained in footnote.

Line 187; Table 5, while there are just (a and b) in the table, no need to write (a-c)!

Line 202: Table 6, Is the mortality percent just for rearing period or from first day till 72 weeks?

Line 208: Table 7, It would be better if the authors measure egg quality at 30 as well because it is almost peak of egg production.

Line 209: Table 7, Again there is no c letter in the table! that is why no need to mention in footnote!

Best regards,

Round 2

Reviewer 1 Report

In my opinion, the article submitted for review can be accepted in its present form

Author Response

Once again, thank you very much for your comments and suggestions.

Reviewer 2 Report

The Authors have strictly followed the suggetions received by the Reviewers, the manuscript can now be accepted for publication.

Author Response

(The authors gave the same response as above.)
